# Connection of Compound Extremes of Air Temperature and Precipitation with Atmospheric Circulation Patterns in Eastern Europe

**Olga Sukhonos** and **Elena Vyshkvarkova** *

Institute of Natural and Technical Systems, 299011 Sevastopol, Russia; kovalenko@instpts.ru
* Correspondence: vyshkvarkova@instpts.ru

**Abstract:** Recent studies show an increase in the frequency of compound extremes in air temperature and precipitation in many parts of the world, especially under dry and hot conditions. Compound extremes have a significant impact on all spheres of human activity, such as health, agriculture, and energy. Features of atmospheric circulation are closely related to the occurrence of anomalies in air temperature and precipitation. The article analyzes the relationship of atmospheric circulation modes with compound extremes that have had the greatest impact on the Atlantic–European region over the territory of Eastern Europe over the past 60 years on extreme air temperature and precipitation. Combinations of extreme temperature and humidity conditions (indices)—cold-dry (CD), cold-wet (CW), warm-dry (WD) and warm-wet (WW)—were used as compound extremes. Indices of compound extremes were calculated according to the E-OBS reanalysis data. Estimates of the relationship between two time series were carried out using standard correlation and composite analyses, as well as cross wavelet analysis. Phase relationships and time intervals for different climatic indices were different. The period of most fluctuations in the indices of compound extremes was from 4 to 12 years and was observed during 1970–2000. The coherent fluctuations in the time series of the WD and WW indices and the North Atlantic oscillation (NAO) index occurred rather in phase, those in the time series of the CD and WD indices and the Arctic oscillation (AO) index occurred in antiphase, and those in the time series of the WD and WW indices and the Scandinavia pattern (SCAND) index occurred in antiphase. Statistically significant increase in the number of warm compound extremes was found for the northern parts of the study region in the winter season with positive NAO and AO phases.

**Keywords:** compound extreme; climate change; circulation mode; Eastern Europe

## 1. Introduction

Global climate change has led to an increase in climate extremes in many regions of the globe [1]. The simultaneous or sequential occurrence of an extreme event related to air temperature and/or precipitation is called a compound extreme [2,3]. In recent years, more and more works have been devoted to the study of compound extremes due to their significant negative impact on human life [4]: on agriculture [5–8], fires [9–11], heat waves and droughts [12–14], public health [15,16], and floods [17].

Studies for different areas and for the world as a whole show a trend towards a decrease in the frequency of cold extremes and an increase in warm ones [18]. This indicates the dominance of the temperature component in the compound extreme [19,20]. Atmospheric circulation is closely related to the occurrence of extreme temperatures and precipitation. Most of the works show the connection of extreme events separately from circulation modes [21–24]. However, in the last decade, works have appeared to reveal the relationship between the occurrence of a compound extremum and atmospheric circulation modes.

Previous studies have identified the NAO as one of the dominant atmospheric patterns influencing the temporal evolution of precipitation and air temperature in the Atlantic–European region [25,26]. However, air temperature and precipitation anomalies are explained not only by the NAO signal, but also by other circulation modes [27].

The following results were obtained in the studies of the European domain. A significant influence of the NAO in winter, as in many parts of the world, was observed for cold-dry and cold-wet conditions in Serbia [28]. Cold-dry and warm-dry conditions were found to be highly negatively and positively correlated with the East Atlantic oscillation (EA) index in all seasons, respectively. In addition, significant positive correlations were obtained between the East Atlantic–West Russia pattern (EA/WR) and cold-dry conditions in autumn and negative correlations were obtained between the former and warm-dry conditions in summer [28]. Using the developed methodology, the authors showed that the peak of the repetition of a compound wet and windy extremum in Europe falls in the boreal winter. High values of co-recurrence ratios correspond to extremely wet and windy events in the northwest of Europe, as well as large-scale conditions resembling the positive phase of the NAO. This confirms earlier findings that link the positive phase of the NAO to an increased frequency of extratropical cyclones impacting northwestern Europe [29].

By dividing the circulation modes into three groups according to the centers of anomalies, Lemus-Canovas [20] obtained estimates of the relationship between compound extremes of air temperature and precipitation in the Mediterranean Basin. The positive phase of the Mediterranean oscillation contributes to increased dryness during warm months in the western Mediterranean Basin, as well as wet-cold extremes in the southeastern Mediterranean Basin. The EA/WR pattern has proven very capable of inferring the occurrence of dry heat and wet-cold phenomena in the eastern Mediterranean [20]. For most mountainous areas of the Mediterranean region, the occurrence of different winter regimes was closely related to the state of the NAO, although the relationship was weaker in the easternmost part of the Mediterranean Basin [26]. Using indices to determine periods of extremely cold or warm temperatures, it was found that teleconnection patterns, such as the NAO, EA, SCAND and EA/WR, in some regions of Europe correlate or anticorrelate to a large extent with cold or heat waves [30].

Evaluation of the relationship between circulation modes and extremes in other regions of the globe shows that the negative phase of the NAO is more likely to cause dry and hot events in Inner Mongolia [31]. For the territory of China, a negative correlation was found between the EA/WR and warm extremes and a positive correlation was found with extreme events of a humid/cold climate for summer. Significant and negative correlations have also been shown between the EA/WR and the spatial extent of complex dry/cold extremes for winter [32]. The relationship between compound dry and hot events and the El Nino–Southern oscillation (ENSO) during the warm season has shown that the likelihood of extremes increases in northern South America, southern Africa, southeast Asia and Australia [33,34]. The ENSO shows a strong association with dry/hot events over the southern hemisphere in summer and autumn, while Pacific decadal oscillation influences their occurrence over western North America in the northern hemisphere during the boreal summer. However, the association of NAO with dry/hot events is relatively weak. The occurrence of dry/hot events in other regions is driven by a combination of these large-scale patterns [35].

The mentioned circulation modes and others are important factors (drivers) of air temperature variability and precipitation in many regions of the globe. Against the background of global warming, the study of spatiotemporal variability of compound extremes of air temperature and precipitation and their relationship with circulation modes remains an urgent task in climatology. Therefore, the purpose of the article is to identify the relationship between compound extreme indices and circulation patterns in Eastern Europe over the past 60 years. To carry this out, along with traditional correlation and composite analyses, we use cross-wavelet analysis to study the relationship between two time series (co-extremum indices and circulation mode indices) in the time–frequency space.

## 2. Materials and Methods

### 2.1. Data

The work used daily data on the average air temperature and the amount of precipitation for the period 1950–2018 for the territory of Eastern Europe (which is geographically central) and northeastern Europe, (25–45° E and 42–61° N, respectively). Data were taken from the E-OBS 20.0 reanalysis (spatial resolution 0.25° × 0.25°) [36]. The E-OBS is a daily gridded land-only observational dataset covering Europe. Quality control of the initial data indicated a sufficient level of data. The amount of data in the grid nodes of the E-OBS reanalysis for the period 1950–2018 exceeded 80% for the entire study area.

### 2.2. Compound Extreme Indices

The compound indices of air temperature (T) and precipitation (P) (Table 1) were used as characteristics of compound extremes. The work used 25th and 75th percentiles in order to detect more events of compound extremes of air temperature and precipitation [37].

**Table 1.** Compound extreme indices.

| Index | Definition | Calculation |
|---|---|---|
| CD | Cold-dry | Days with T < 25th percentile and P < 25th percentile |
| CW | Cold-wet | Days with T < 25th percentile and P > 75th percentile |
| WD | Warm-dry | Days with T > 75th percentile and P < 25th percentile |
| WW | Warm-wet | Days with T > 75th percentile and P > 75th percentile |

The threshold values of air temperature and precipitation were calculated for each season for the base climatic period 1961–1990. The compound temperature and precipitation extremum were determined in the case of the coincidence of the corresponding extreme on a specific day for the period 1950–2018. Further, the total number of such coincidences per month/season/year was determined. It should be noted that the percentiles in the precipitation series were calculated from the time series with days when precipitation exceeded 1 mm. The winter season (DJF) corresponds to January and February of the calendar year and December of the previous year.

### 2.3. Atmospheric Circulation Modes

To analyze the relationship between the indices of compound extremes and circulation modes, the indices of the North Atlantic Oscillation, East Atlantic Oscillation, Scandinavian Oscillation, Arctic Oscillation and East Atlantic–West Russia pattern for the period 1950–2018 were used. The indices are taken from the Climate Explorer website (https://climexp.knmi.nl, accessed on 8 January 2023).

The NAO is the leading circulation mode in the North Atlantic [38], and phase changes lead to significant changes in atmospheric circulation in the Atlantic–European region [39,40]. The NAO has the greatest impact on weather and climate conditions in Europe, especially in winter [41–43], which is associated with changes in cyclone trajectories in the Atlantic–European region [44].

The East Atlantic pattern is the second mode of low frequency variability over the North Atlantic. The EA centers anomalies are shifted to the southeast relative to the centers of the NAO [45]. The positive phase of EA is associated with higher average surface temperatures in Europe in all months, above average precipitation in northern Europe and below average precipitation in southern Europe [45].

The Arctic Oscillation refers to a pattern of atmospheric circulation over the middle and high latitudes of the Northern Hemisphere [46]. The AO's positive phase is characterized by below-average air pressure over the Arctic combined with above-average air pressure over the North Pacific and Atlantic oceans. Thus, in the middle latitudes of North America, Europe, Siberia, and East Asia, fewer flashes of cold air are usually observed in the positive AO phase than usual. Conversely, the negative phase of the AO has an atmospheric

pressure that is above the average over the Arctic region and that is below the average over the northern part of the Pacific and Atlantic oceans. Areas at mid-latitudes are more likely to experience flashes of cold polar air in winter when the AO is negative [47].

The Scandinavia pattern is associated with anomalies in heights over Scandinavia and western Russia [45]. During the SCAND's positive phase, temperatures are below average in central Russia and western Europe, and precipitation is above average in central and southern Europe. These positive and negative precipitation anomalies are in good correspondence with intensified and reduced storm-track activity [45,48].

The EA/WR model [45] is one of three known telecommunication models that affect Eurasia throughout the year. The EA/WR mode is characterized by two main large-scale anomalies located over the Caspian Sea and Western Europe [45], and also reaches the territory of the Middle East [49]. The positive and negative EA/WR phases create positive and negative temperature anomalies, respectively, over the eastern United States, Western Europe and Russia east of the Caspian Sea, with negative and positive anomalies, respectively, over eastern Canada, eastern Europe including the Ural Mountains, northeast Africa and Central Asia. Positive and negative precipitation anomalies are found over the mid-latitude Atlantic and the central part of Russia around ~60° E, where the low-level cyclonic and anticyclonic circulation anomalies, respectively, predominate. Eastern Canada and Western Europe, including the Mediterranean region, are characterized by negative and positive precipitation anomalies, respectively [50].

*2.4. Statistical Analysis*

To identify the relationship between the indices of compound extremes and circulation modes, the Pearson correlation coefficient was calculated and its statistical significance was determined by Student's *t*-test at the 95% significance level.

In terms of composite analysis, to isolate the positive and negative phases of the circulation mode, the index of a specific mode was ranked and 20% of the length of the time series was selected from each side (in our case, 14 values). That is, we selected years with the maximum absolute values of the index for each phase, positive or negative. Next, a comparison was made of the number of days with the compound extreme index in the positive and negative mode phases. The determination of the statistical significance of the difference between the values of the index in the positive and negative phases of the circulation mode was evaluated by Student's *t*-test (95% significance level).

The relationship between two time series in a time–frequency space has been studied by cross wavelet transform and wavelet coherence analyses (introduced by the Torrence and Compo [51]).

The cross wavelet transform (XWT) of two time series, $x_n$ and $y_n$, is defined as

$$W_n^{xy} = W_n^x W_n^{y^*} \qquad (1)$$

The cross wavelet spectral density can be defined as |Wxy|. The XWT reflects the high-energy region of the two time series and finds the region of consistent periodic intensity in the time series in the time–frequency space, which generally reflects the common cycle intensity between the series.

Wavelet coherence can be used to quantify the strength of covariance between two signals, finds the region in time–frequency space where two time series vary together and focuses more on the correlation between two time series in the low-energy region [52].

Wavelet coherence is computed as [53]

$$R_n^2(S) = \frac{\left| S(s^{-1} W_n^{xy}(s)) \right|^2}{S\left( s^{-1} |W_n^x(s)|^2 \right) \cdot S\left( s^{-1} \left| W_n^y(s) \right|^2 \right)}, \qquad (2)$$

where *S* is the smoothing operator.

Overall, wavelet coherence can be thought of as the local, or time-resolved correlation between two time series [51,54].

## 3. Results

### 3.1. Frequency of Compound Extremes

The average number of CD (cold-dry) days in winter for the study region is 1–3 days a year (Figure 1). The number increases to 4 days on the eastern coast of the Black Sea (the Black Sea coast of the Caucasus). The CW (cold-wet) combination occurred up to 1–2 days a year in the entire region, except the Caucasus (up to 5 days a year). The WD (warm-dry) combination amounts to 2–3 days a year up to the 52nd parallel, while northwards it increases to 3–4 days a year. The WW (warm-wet) days are the most numerous in the region during winter. For the south of European Russia, eastern and Northern Ukraine, and Belarus, the number increases to 5 days a year, and in the north of the study region, this number of days with this index is 7–8 per year.

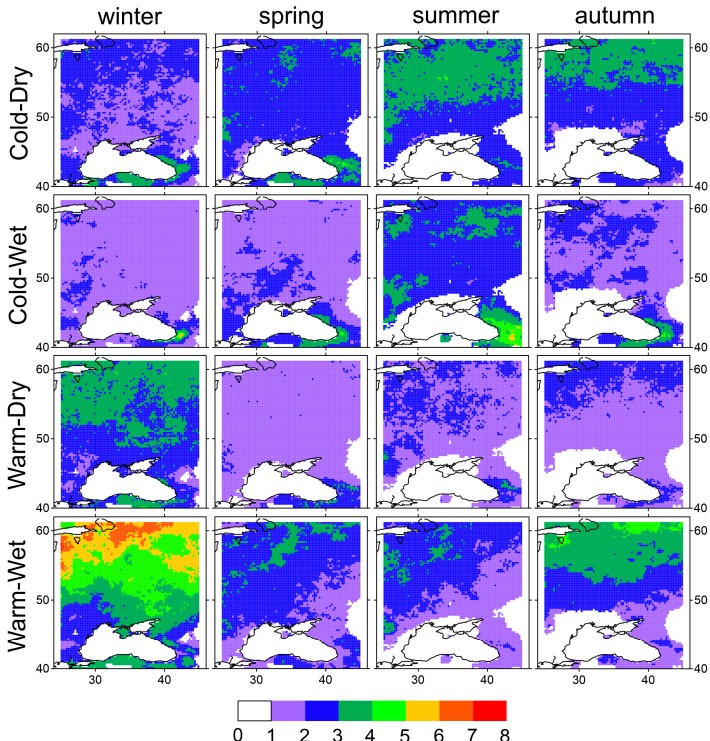

**Figure 1.** Average number of days per year with compound extreme indices for the period 1950–2018 over the Eastern Europe.

The number of days per year with the compound air temperature and precipitation extremes in spring is lower than that in winter. All of the indices occur 1–3 days a year. The CD occurs more frequently in the Caucasus (up to 5 days a year), the CW is often observed on the eastern coast of the Black Sea (up to 5 days a year), and the WW occurs in the northern parts of the region.

In summer, the number of compound CD extremes to the north of the 52th parallel reaches 3–4 days, and to the south it does not exceed 2–3 days per season per year. The CW index in most of the study region is 2–3 days, and an increase in the frequency of compound extremes is observed in the region of the Caucasus Range (up to 6 days). The frequency of the WD throughout the region does not exceed 3 days per year and has a mixed structure. The frequency of the WW in the northwest of the region reaches 4 days per year per season, and in the southeast it reaches no more than 2 days.

In the autumn season, the frequency distribution of the CD index is similar to that in the summer season. The frequency of the CW does not exceed 3 days per year throughout the region, with a slight increase on the Black Sea coast of the Caucasus (up to 4–5 days). The repeatability of the WD is 1–2 days for almost the entire region, and only in the north of the region there is a slight increase to 3 days. The frequency of the WW is characterized by a zonal structure—to the south of the 49th parallel, the number of days with the index does not exceed 2 days, and to the north, the number of days gradually increases and in the north of the region reaches 5 days.

### 3.2. Impacts of Atmospheric Circulation Modes on the Compound Extreme Indices

The correlation coefficients averaged for the entire study region between the indices of compound extremes and circulation modes are shown in Table 2. The CD index has a statistically significant negative relationship with the NAO, AO, and EA circulation modes in the winter season. In spring, a significant negative relationship was found between the CD and the NAO, and in summer, the EA mode. A positive and statistically significant relationship between cold-dry conditions and the EA/WR mode was obtained for summer. The autumn season is characterized by a statistically significant negative relationship with the EA and SCAND, and by a positive relationship with the EA/WR. The second cold index of CW compound extremes has a significant negative relationship with the NAO and AO in winter and spring seasons, with the EA in summer and the NAO in autumn. In summer, a positive significant relationship was found between the CW index and the EA/WR mode. Warm indices (WD and WW) are characterized by the opposite relationship. In winter, a positive relationship was found between the warm indices and the NAO, EA, and AO modes, and a negative relationship was found with the EA/WR mode (for all seasons). Negative correlation coefficients were obtained for warm indices and for the SCAND mode in the winter and autumn seasons.

**Table 2.** Correlation coefficients between the number of compound extreme indices and the circulation modes averaged for the study regions; 25–45° E and 42–61° N.

| Index | Circulation Mode | Season | | | |
|-------|------------------|--------|--------|--------|--------|
| | | **Winter** | **Spring** | **Summer** | **Autumn** |
| CD | NAO | **−0.49** | **−0.27** | 0.12 | −0.14 |
| | EA | **−0.27** | −0.13 | **−0.40** | **−0.29** |
| | AO | **−0.35** | −0.23 | −0.02 | 0.08 |
| | SCAND | 0.14 | 0.14 | −0.03 | **−0.36** |
| | EA/WR | 0.23 | 0.05 | **0.38** | **0.36** |
| CW | NAO | **−0.40** | **−0.29** | −0.02 | **−0.38** |
| | EA | −0.06 | 0.16 | **−0.32** | −0.08 |
| | AO | **−0.35** | **−0.38** | −0.18 | −0.16 |
| | SCAND | 0.24 | 0.01 | −0.02 | −0.21 |
| | EA/WR | 0.07 | 0.05 | **0.31** | 0 |
| WD | NAO | **0.61** | −0.10 | −0.07 | 0.07 |
| | EA | **0.31** | 0.15 | **0.49** | 0.06 |
| | AO | **0.61** | 0.08 | 0.13 | 0.23 |
| | SCAND | **−0.34** | −0.09 | −0.07 | **−0.29** |
| | EA/WR | 0.03 | **−0.32** | **−0.54** | **−0.28** |
| WW | NAO | **0.51** | −0.13 | −0.16 | **−0.28** |
| | EA | **0.29** | **0.27** | **0.54** | **0.33** |
| | AO | **0.27** | −0.02 | 0.09 | 0 |
| | SCAND | **−0.32** | −0.15 | −0.12 | −0.11 |
| | EA/WR | −0.21 | **−0.38** | **−0.61** | **−0.38** |

Bold—statistically significant correlation coefficient ($p < 0.05$).

The spatial distribution of the difference between the average values of the indices of compound extremes between different phases of the circulation modes is shown in Figure 2. Cold extremes (the CD and CW indices) are more common in Eastern Europe during the negative NAO phase. The change in the number of days with indices of compound extremes between different phases of the AO in the winter season is similar to that between different phases of the NAO. In the summer season, when the EA is negative, the number of days with the CD index is higher throughout the region, and in the north of the region it is statistically significant. The SCAND is clearly manifested in the autumn season by an increase in the number of CD compound extremes in the negative phase. The number of days with CW conditions has a heterogenous structure with small areas of statistically significant differences between the phases of the circulation modes. A significant increase in the number of days with CW during the positive SCAND phase was observed on the western coast of the Black Sea, while during the positive EA/WR phase this was observed on the eastern coast of the Black Sea (Black Sea coast of the Caucasus).

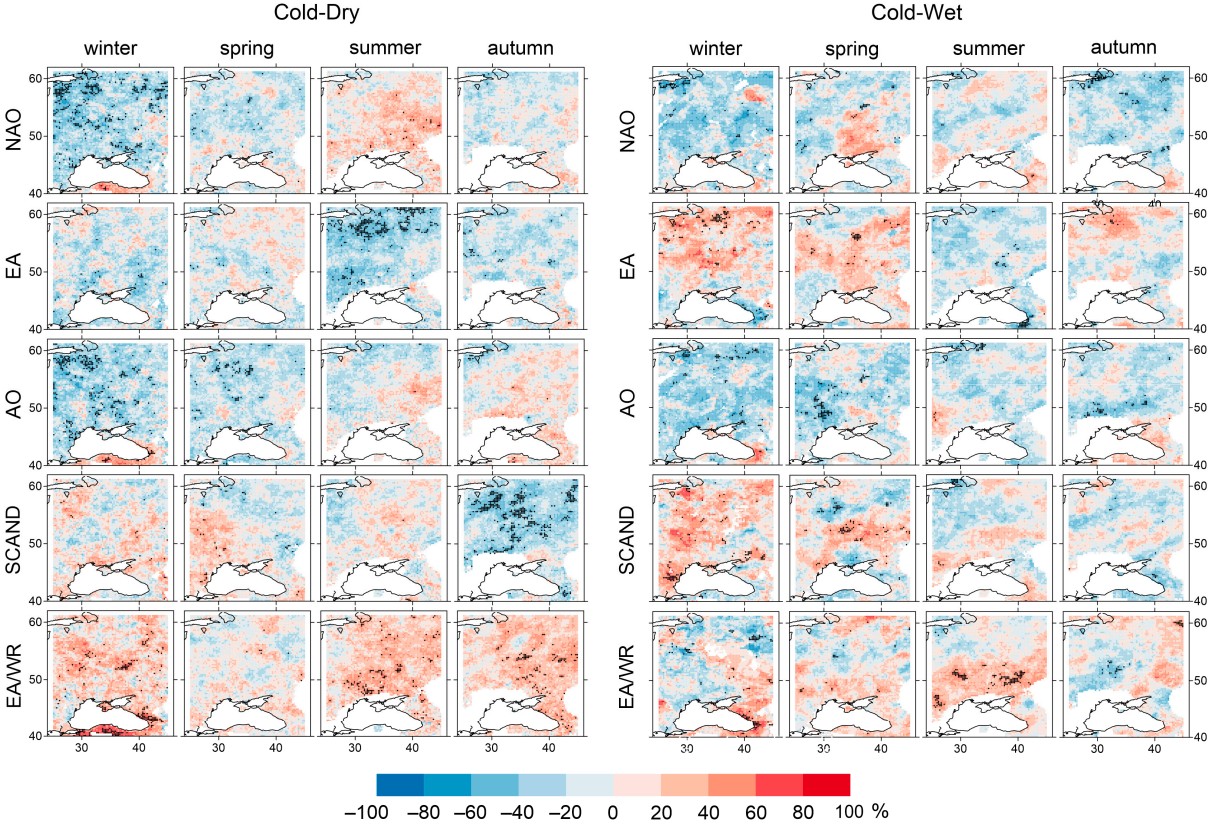

**Figure 2.** Composites maps of the differences of average values (in %) of cold compound extreme indices (CD and CW) between the positive and negative phases of the circulation mode for the period 1950–2018. Black dots—statistically significant difference ($p < 0.05$).

The manifestations of circulation modes in the variability of warm extremes are more pronounced than those in cold ones (Figure 3). Statistically significant differences between the number of warm compound extremes in different phases of circulation modes were found mainly for the winter and summer seasons. The WD and WW indices have higher values during the positive NAO and AO phase in the north of the study region during the winter season (the difference reaches 100%), and during the negative SCAND phase. The number of the WW compound extremes is higher in the positive EA phase and in the negative EA/WR phase in the summer season throughout the region. In the winter season, in the region of the Caucasus Ridge, when the NAO and especially the AO are in the negative phase, the number of days with warm indices is higher than that in the positive one.

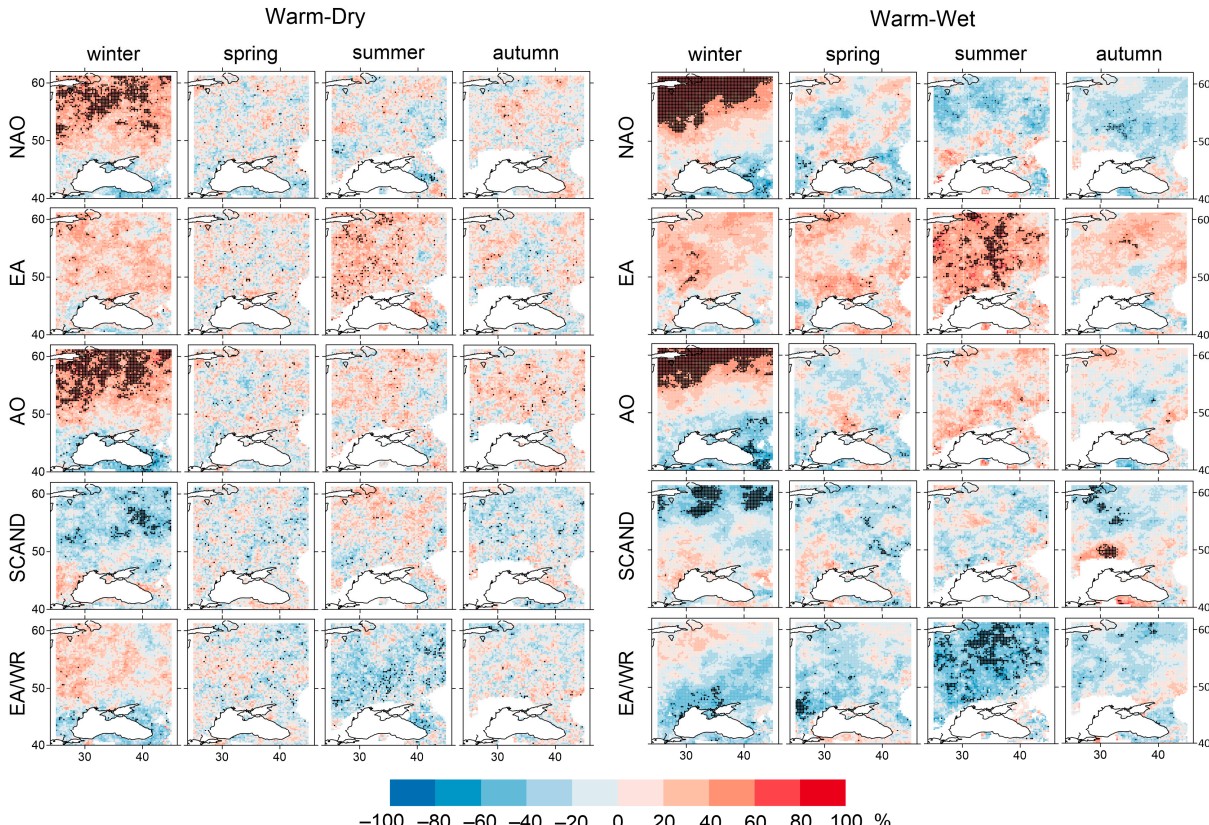

**Figure 3.** Composites maps of the differences of average values (in %) of warm compound extreme indices (WD and WW) between the positive and negative phases of the circulation mode for the period 1950–2018. Black dots—statistically significant difference (*p* < 0.05).

An analysis of difference composites showed statistically significant differences in the number of days with compound extreme indices in different phases of circulation modes in winter and summer seasons. Therefore, the consistency of fluctuations in the series of compound indices with climate indices was studied using cross-wavelet analysis for these seasons. Regions in which coherent oscillations were observed with a time interval comparable to the period of oscillations were not considered. The power spectrum is shown in Figure 4.

For winter, it was found that the WD index has significant coinciding periods in the range of 7–12 years with the NAO, SCAND, and AO during the period 1972–2008 (Figure 4). At the same time, the WD index positively correlates with the NAO and AO indices and negatively correlates with the SCAND index. The CD index has significant overlapping periods in the range of 8–12 years with the NAO, SCAND, AO during the period 1969–1994, is negatively correlated with the NAO, AO and is positively correlated with the SCAND. It should be noted that the NAO in the period 1976–1986 lagged behind the CD index by 1–1.5 years; the AO until 1979 lagged by 1–1.5 years, and after 1979 the climate index and the CD index changed in phase. The cross-wavelet spectrum between the CW, WW and mode indices show small areas with coherent fluctuations. It can only be noted that the WW index has significant coinciding periods in the range of 4–8 years with the EA/WR during 1992–2002; a significant negative correlation is noted between the EA/WR and the considered index with a delay of 0.5–1 year. Consistency of fluctuations in the time series of compound indices with the EA was not found throughout the analyzed period.

Cross-wavelet coherence analysis showed significantly high and positive correlations between the WD index and the NAO during 1957–2009 in periods of 4–8 years and 8–16 years (Figure 5). At the same time, in the period of 4–8 years, the NAO was ahead of the WD index until 1971, and since 1993 it has lagged behind; in a period of 8–16 years,

coherent fluctuations between the indices were in-phase during 1963–1990. Significant negative correlations were found between the SCAND and the WD index in a period of 2–4 years, 4–8 years, and 8–16 years. The SCAND lagged behind the considered index by 0.25–0.5 years during 1983–1991 in a period of 2–4 years, by 0.75–3 years in periods of 4–8 years during 1984–1995 and by 8–16 years during 1981–2004. Positive correlations between the EA/WR and the WD index were observed in a period of 16–20 years during 1973–1999; the EA/WR was ahead of the WD index by 2–2.5 years. Significantly high and positive correlations between the WD index and the AO were revealed in periods of 4–8, 8–16, 16–20 years during 1964–2009; the AO led the WD index by 1.5–3 years during 1981–1996, and by 1–2 years during 1967–1981.

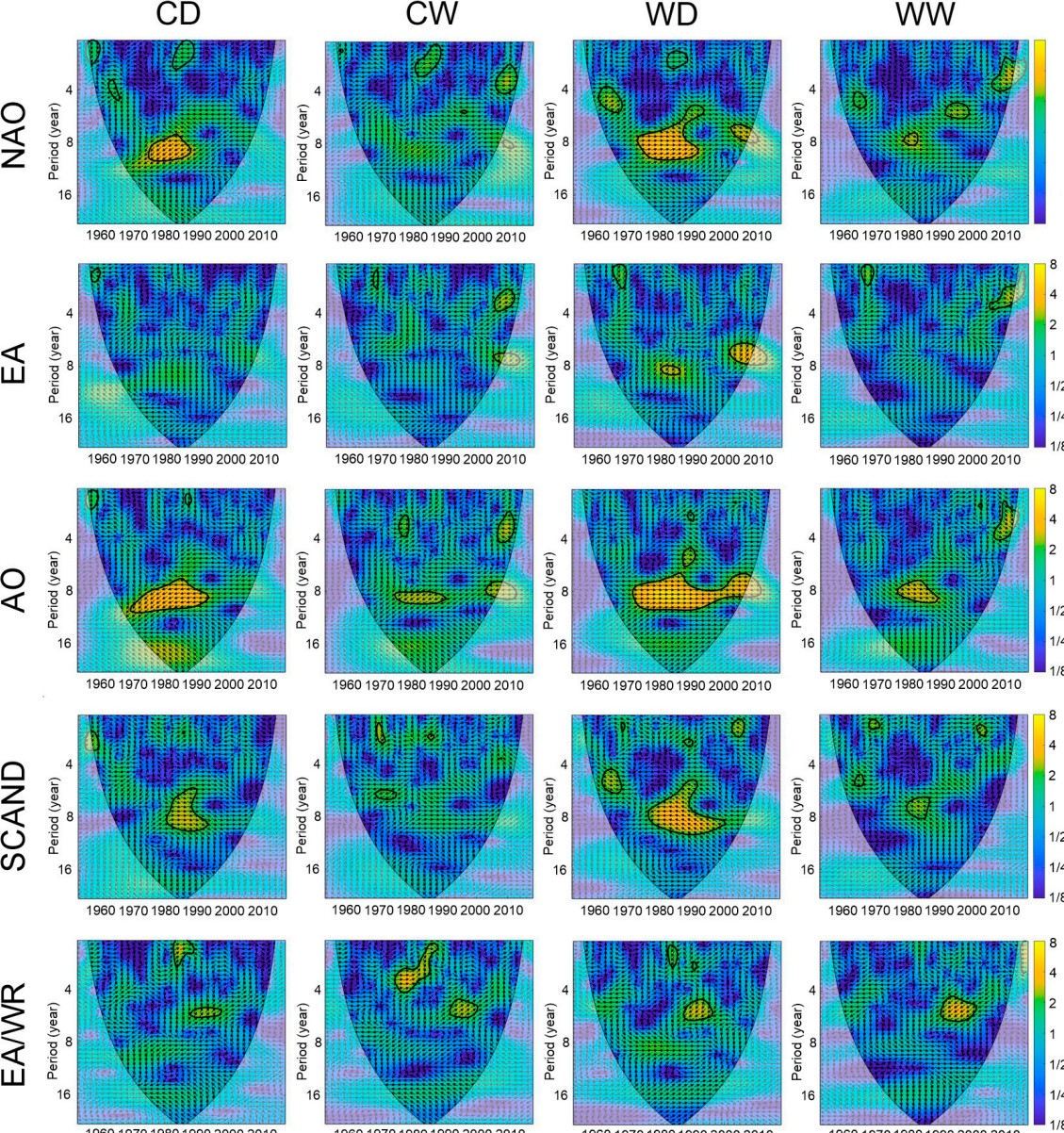

**Figure 4.** Cross-wavelet spectrum between compound extreme indices and circulation mode for winter season. Thick black contour lines determine the 5% significance level for red noise. Cones of influence (COI), pictures that may distort the edges, are represented in lighter tones. Right-pointing arrows: in-phase relationship. Left-pointing arrows: antiphase relationship. Down-pointing arrows show that the climate index leads the compound extreme index by 90° (one-quarter). Yellow represents stronger power and blue represents weaker power.

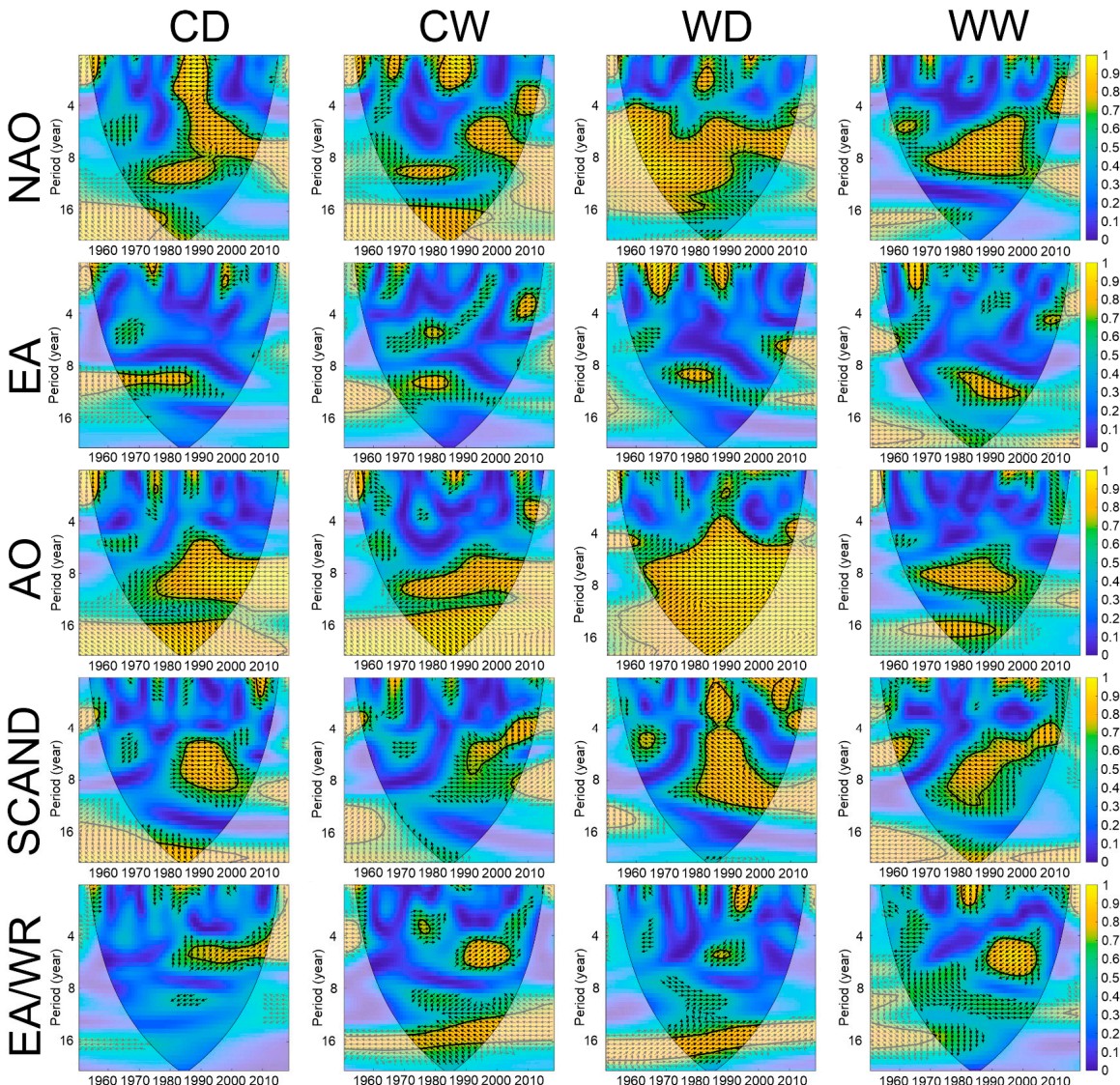

**Figure 5.** Wavelet coherence spectrum between compound extreme indices and circulation modes for winter season. Thick black contour lines delineate the 5% significance level for red noise. COI pictures that may distort the edges, are represented in brighter tones. Right-pointing arrows show an in-phase relationship, left-pointing arrows show an antiphase relationship and down-pointing arrows show that the atmospheric mode index leads to a compound extreme index 90° (one-quarter). Yellow represents stronger power and blue represents weaker power.

Negative correlations between the CD index and the NAO are observed on the range of periods from 2 to16 years during 1973–2007 and between the CD index and the AO index in the period of 6–12 years during 1977–2007 and in the period of 16–20 years during 1975–1997. At the same time, the NAO lagged behind the considered index by 1–2 years for a period of 8–16 years during 1973–1994, and the AO index lagged by 6–7.5 years during 1975–1997. The CD index is positively correlated with the EA index over a period of 8–16 years during 1965–1985, and with the SCAND index over a period of 5–10 years during 1985–2001. It should be noted that the EA index is ahead of the CD index by 2–4 years. Positive correlations are also noted between the CD index and the EA/WR index for a period of 4–8 years during 1989–2012; the EA/WR is 1–2 years ahead.

The largest areas of negative correlations between the NAO, AO and CW indices are observed in a period of 6–8 years during 1971–2007 and in a period of 16–20 years during 1973–1998. The areas of correlations between the EA and the considered index

are insignificant. Over a period of 4–8 years, the CW index is positively correlated with the SCAND index during 1992–2004, and negatively correlated with it during 2005–2009, lagging by 1–2 years. The CW index is positively correlated with the EA/WR index over a period of 4–8 years during 1991–2003 (leading by 0.5–1 year) and negatively correlated with it over a period of 14–18 years during 1973–2000 (with a lag of ~5–6 years).

The WW index has positive correlations with the NAO over a period of 6–10 years during 1969–1999 and with the AO over a period of 6–10 years during 1968–1997. In the spectrum of coherence between the WW index and the AO, there is also an area of negative correlations over a period of 16–20 years during 1971–1992; the AO is 4–5 years late. With a lag of 1–2 years, the EA index is negatively correlated with WW index over a period of 8–16 years during 1981–1997. The considered index is negatively correlated with the SCAND index for a period of 4–12 years during 1997–2002, and with the EA/WR over a period of 4–7 years during 1989–2003 (with a delay of ~0.5 years). At the same time, the area of positive correlations between the WW index and the SCAND index in the period of 4–6 years during 2003–2009 is highlighted (with a lead of 1–1.5 years).

Significant regions are predominantly absent in the coherence spectrum for the summer season, which generally indicates the absence of a significant relationship between the time series of circulation modes and compound extreme indices over the territory of Eastern Europe (Figures S1 and S2, Supplementary Files).

## 4. Discussion

The continuous increase in air temperature contributes to an increase in the frequency of extremes in many parts of the globe. Previously, it was shown that negative trends for cold indices were found in Eastern Europe, and positive trends were found for warm ones [55]. Similar trends are typical for most regions of the globe [56–58]. Model calculations of the CMIP6 project predict an increase in the frequency of combined droughts and hot weather (compound droughts and hot events) both hydrological and agricultural throughout the 21st century [59–61]. Li et al. [62] predict an increase in the number of people who will be overstressed amid an increasing frequency of compound extreme heat and dry conditions (heat waves) across the globe. Similar results were obtained by Wu et al. [63], Ma and Yuan [64], who showed an increase in warm summer extremes in most parts of the world under different representative concentration pathway and shared socioeconomic pathway scenarios.

Our results show a pronounced seasonal variability in the relationship between the compound extreme indices of air temperature and precipitation and atmospheric circulation modes. According to the analysis of difference composites, the atmospheric circulation modes have the greatest influence on the number of compound extreme indices in the winter and summer seasons in the northern parts of the region. An increase in the number of cold and warm extremes occurs in opposite phases of the circulation modes. In the positive phase of the NAO and AO modes, an increase in the number of warm extremes in the winter season was found. Many works show that the NAO determines the extreme values of air temperature, precipitation, wind and snow cover in Europe, which are especially pronounced in the winter season [65–67]. In the winter season, cold extremes and the NAO, EA, and AO modes have a statistically significant negative correlation. Negative correlations between these parameters were also obtained for other regions of Europe [28]. With a high NAO index over parts of southern Europe, there is a significant decrease in atmospheric moisture transport [28,38]. The relationship between compound extreme indices and large-scale circulation patterns showed that EA and NAO had a significant impact on the duration of winter warm periods, while their influence on the duration of cold periods could not be reliably confirmed [56]. Ionita et al. [68], analyzing the standardized precipitation—evapotranspiration index, found that in winter, the dominant mode of variability in aridity and humidity is influenced by the AO, NAO, SCAND and EA/WR.

In summer, we obtained a positive statistically significant relationship between the warm indices of compound extremes and the EA index. With a positive EA phase, zonal circulation of the atmosphere prevails over Europe [66,69]. This is reflected in the storm-tracks and affects the surface air temperature (which becomes above normal) and precipitation in the region [70,71]. For extremely warm and humid conditions (WW), a statistically significant negative relationship was obtained with the EA/WR index in the summer season, as in [72] for the period 1976–2019. In spring and autumn, it was not possible to obtain clear results for most of the signals, with rare exceptions. For example, the CD index has a greater number of days with a negative SCAND phase almost throughout the region, and in the northern parts of the region, the difference is statistically significant. For the WW index, one should single out an increase in the number of days with the index in the negative phase of the EA/WR in spring near the western coast of the Black Sea. A statistically significant increase in the number of days with the WW index in the north of the region is characteristic of the negative SCAND phase in autumn, while in the central region of the region there is an area with a statistically significant increase in the frequency of the index in the positive phase of the SCAND pattern.

The manifestations of the NAO and AO in the indices of compound extremes over the territory of Eastern Europe are similar. These two modes are strongly correlated with each other [73], and it is believed that the NAO is a regional manifestation of the AO [74]. As in the case of the NAO, the AO is the main source of intraseasonal variability over the North Atlantic and Europe in winter, but its influence remains noticeable in other seasons [24]. Warm indices (the WD and WW) in the winter season showed a statistically significant positive relationship, that is, a significant increase in the number of days with compound extremes in the positive AO phase. After analyzing the relationship between atmospheric circulation indices and temperature fluctuations in Russia over the period 1976–2019, Perevedentsev et al. [72] found that the AO and NAO have a significant effect on the thermal regime in winter in the European part of Russia. Overland and Wang [14], using the example of a heat wave in Siberia from January to April 2020, showed the influence of the positive AO phase, which is an expression of the strength of the stratospheric polar vortex, on air temperature anomalies over Eurasia. Tabari and Willems [75] showed a dipole-type pattern of winter extreme precipitation over Europe: a significant positive correlation with the NAO in the north and west of Europe, and negative for southern Europe. A similar result was obtained for the AO but relatively weaker.

Joint periodicities in the time series of different hydrometeorological characteristics for the analysis of regional manifestations of climate change are often studied using cross-wavelet transform and wavelet coherence analysis [22,76,77]. It is observed that these circulation modes had the greatest impact on the indices of compound extremes in the study region during the winter season. We found that fluctuations in the NAO and AO were positively and negatively correlated with the appearance of warm-dry and cold-dry conditions, respectively. This obtained high correlation of compound extremes with the NAO and AO fluctuations of similar frequency is coherent over a wide range of scales from 4 to 16 years. It is consistent with the fact that during a pronounced positive phase of the NAO, above-average air temperatures and precipitation are detected in northern Europe in the winter season [45]. The SCAND changes are coherent in the range from 4 to 10 years with compound extremes being similar in frequency, correlating negatively with the appearance of warm-dry conditions and correlating positively with the appearance of cold-dry conditions. The EA/WR fluctuations are coherent in the range of 4–8 years with compound extremes that are similar in frequency and that correlate negatively with the appearance of warm-wet conditions.

The period of most fluctuations in the compound extreme indices is from 4 to 12 years and is observed during 1970–2000. Coherent fluctuations in the time series of warm indices (the WD and WW) and the NAO index occurred rather in phase, those in the time series of CD and WD indices and the AO index occurred in antiphase, and those in the series of WD and WW indices and the SCAND index occurred in antiphase. For different locations in

Europe, cross-spectral analysis between precipitation and winter NAO time series revealed the best coherence in a dominant cycle between 3 and 4 years [78].

Despite the presence of statistically significant differences in the number of compound extremes between the phases of circulation modes in the summer season, no significant regions were found in the coherence spectrum, which indicates, in general, that there is no significant relationship between the time series of circulation modes and the indices of compound extremes over the territory of Eastern Europe in the warm season. An anticyclone is set up for a long time over the study region in summer, which determines hot and dry conditions, especially in the southern regions [1,79].

## 5. Conclusions

Using five atmospheric circulation modes that have the greatest influence on the territory of the Atlantic–European region, we examined the change in the compound extremes of air temperature and precipitation in different phases of these modes. The analysis was carried out for four compound extreme indices—two cold indices (the CD and CW) and two warm indices (the WD and WW)—for the period 1950–2018 over Eastern Europe. Our findings can be summarized as follows:

1.  The circulation modes play an important role in the dynamics of the compound extreme index regimes and showed pronounced seasonal variability in the relationship between them.
2.  The atmospheric circulation modes have the greatest influence on the number of compound extreme indices in the winter and summer seasons in the northern parts of the considered region. An increase in the number of cold and warm extremes occurs in opposite phases of the circulation modes.
3.  The AO and NAO more strongly affect the warm compound extreme indices in winter (positive correlation), while the EA/WR pattern in summer has a greater effect on the region with a negative correlation.
4.  The NAO, SCAND and AO have significant coinciding periods with the warm compound extreme indices in the range of 7–12 years. The EA/WR has significant coinciding periods with all compound extreme indices in the range of 4–8 years.

In addition to analyzing the connection of compound extremes with circulation modes separately, it is important to analyze their combined influence on the occurrence of air temperature and precipitation anomalies, since they influence each other [39,80–82]. Comas-Bru and McDermott [66] showed that combined NAO and EA analyses more accurately describe winter climate variability in Europe. The study of the relationship between compound extremes and circulation modes creates the basis for predicting these phenomena and understanding the nature of their occurrence.

**Supplementary Materials:** The following supporting information can be downloaded at: https://www.mdpi.com/article/10.3390/cli11050098/s1, Figure S1. Cross wavelet spectrum between compound extreme indices and circulation mode in-dices for summer season; Figure S2. Wavelet coherence spectrum between compound extreme indices and circulation modes for summer season

**Author Contributions:** Conceptualization, O.S. and E.V.; methodology, O.S.; formal analysis, O.S.; investigation, O.S.; data curation, O.S.; writing—original draft preparation, O.S. and E.V.; writing—review and editing, O.S. and E.V.; visualization, E.V.; supervision, E.V. All authors have read and agreed to the published version of the manuscript.

**Funding:** The study was supported by state assignment of Institute of natural and technical systems (project reg. no. 121122300072-3).

**Data Availability Statement:** The initial time series of daily data of the average air temperature, precipitation and circulation patterns indices are on the website Climate Explorer (European Climate Assessment & Dataset) https://climexp.knmi.nl/start.cgi (accessed on 8 January 2023).

**Acknowledgments:** The authors are grateful to the anonymous reviewers for the remarks and comments which led to the improvement of the paper.

**Conflicts of Interest:** The authors declare no conflict of interest.

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
