# Peer review of "Connection of Compound Extremes of Air Temperature and Precipitation with Atmospheric Circulation Patterns in Eastern Europe"

_climate, doi:10.3390/cli11050098_

Round 1

Reviewer 1 Report

Specific comments

 Grammar and English notes:

Line 100:What are E-obs ?  Please give the full names when they first appear. Similarly, in line 300:COI; Line 368:RCP, SSP; Line 380: SPEI

Line 103,” the period 1950-2018. exceeds 80% ”, there is a full stop after 2018.

Line 106-107:”The work used not so-called "extreme" percentiles (25th and 75th) in order to detect more…”,  but in fact the compound extremes index is defined using percentile thresholds as shown in Table. 1.

line 108: [37[.   The right square bracket ] is used incorrectly.

line 124. Please add accessed date.

line 130: “anomalyies” should be “anomalies”

Line 205:”is lower than in winter”,it should be added “that” after word “than”. Similarly, in line 261,269.

Line 208,214,216,217,219,221,the word “index” should be deleted.

line 284,287:SKAND should be SCAND

Line 326:”is 1–2 years ahead”, after the word “ahead”, lack of a full stop.

Line 330: WC should be CW

Line 340:SCAN should be SCAND

Line389: WD should be WW,the features that are described in line 389-395 are WW’s features not WD’s.

Figures, data, methods and analysis notes:

All the figures are not clear. line 256: The longitude and latitude are not given for the horizontal and vertical coordinates in Figure 2. line 273. Figure 2 should be changed to Figure 3.

Line 115-117: “. It should be noted that the percentiles in the precipitation series were calculated from the time series with days when precipitation exceeded 1 mm.” The author just sorted and calculated the index from the time series with days when precipitation exceeded 1 mm, and did not distinguish between snowfall and precipitation in winter.

line 225: What does white color represents in Figure 1. Please depict in the legend.

Line 224-226: As can be seen in Figure 1, except WW in winter, the compound extremes index values in other seasons are all small. Is the statistics of this composite index representative?

There is little analysis of CW in Figure 2.

The values in figure 2 and 3 is given in %. How is it calculated? The correlation coefficient given in Table 2 does not indicate whether it is the regional average of the correlation coefficient or the correlation coefficient after regional average.

The methods and data are not well described. In the case of global warming, as you can see in Figure 1, the warm and wet extreme index are robust in the winter and the differences of Warm events (WW and WD) in winter between the positive and negative of NAO and AO are significant. The author does not describe how the data is processed,whether the trend is removed from the original data.

All of the analysis describes the characteristics of the Figures or the relevant results in the table, but does not analyze the mechanism.

line 409. Discussion is too short and mixed with conclusions. It is suggested to separate the summary from the discussion. I advise to expand discussion section containing comparison of the results with other studies.

It is meaningful to obtain the period, phase difference and correlation by wavelet spectrum analysis and coherence analysis, but it should be discussed in the sense of weather and climate.

Author Response

Response to Reviewer 1 Comments

Dear Reviewer!

First of all, we want to thank you the reading our article carefully and for your valuable comments! Below are the answers to your comments

Grammar and English notes:

Point 1: Line 100:What are E-obs ?  Please give the full names when they first appear. Similarly, in line 300:COI; Line 368:RCP, SSP; Line 380: SPEI

Response 1: Full names are given after the first mention

Point 2: Line 103,” the period 1950-2018. exceeds 80% ”, there is a full stop after 2018.

Response 2: Corrected

Point 3: Line 106-107:”The work used not so-called "extreme" percentiles (25th and 75th) in order to detect more…”,  but in fact the compound extremes index is defined using percentile thresholds as shown in Table. 1.

Response 3: The sentences is rewritten

Point 4: line 108: [37[.   The right square bracket ] is used incorrectly.

Response 4: Corrected

Point 5: line 124. Please add accessed date.

Response 5: Accessed date added

Point 6: line 130: “anomalyies” should be “anomalies”

Response 6: Corrected

Point 7: Line 205:”is lower than in winter”,it should be added “that” after word “than”. Similarly, in line 261,269.

Response 7: Corrected

Point 8: Line 208,214,216,217,219,221,the word “index” should be deleted.

Response 8: Corrected

Point 9: line 284,287:SKAND should be SCAND

Response 9: Corrected

Point 10: Line 326:”is 1–2 years ahead”, after the word “ahead”, lack of a full stop.

Response 10: Corrected

Point 11: Line 330: WC should be CW

Response 11: Corrected

Point 12: Line 340:SCAN should be SCAND

Response 12: Corrected

Point 13: Line389: WD should be WW,the features that are described in line 389-395 are WW’s features not WD’s.

Response 13: You are right, there should be WW here. Thank you!

Figures, data, methods and analysis notes:

Point 14: All the figures are not clear.

Response 14: The longitude and latitude are added to the all figures. The font on the pictures is enlarged

Point 15: line 256: The longitude and latitude are not given for the horizontal and vertical coordinates in Figure 2.

Response 15: The longitude and latitude are added to the Figure 2 and on all others

Point 16: line 273. Figure 2 should be changed to Figure 3.

Response 16: Corrected

Point 17: Line 115-117: “. It should be noted that the percentiles in the precipitation series were calculated from the time series with days when precipitation exceeded 1 mm.” The author just sorted and calculated the index from the time series with days when precipitation exceeded 1 mm, and did not distinguish between snowfall and precipitation in winter.

Response 17: That's right, we don't separate winter precipitation into snow and liquid precipitation. We use daily precipitation totals (reduced to the thickness of the water layer in millimeters). It is a common practice to use precipitation when there is no need to study snow cover. Thanks for the comment, in the following works we will try to make such a separation.

Point 18: line 225: What does white color represents in Figure 1. Please depict in the legend.

Response 18: The legend in the Figure 1 is depicted. White color represents zero index value

Point 19: Line 224-226: As can be seen in Figure 1, except WW in winter, the compound extremes index values in other seasons are all small. Is the statistics of this composite index representative?

Response 19: The maps show the average annual values of the index of joint extremes. In some years the values are really small, and in some years it reaches 15 or more values per year. Spatial maps do not show the change in indices over time.

Point 20: There is little analysis of CW in Figure 2.

Response 20: Description of index CW results added to manuscript

Point 21: The values in figure 2 and 3 is given in %. How is it calculated?

Response 21: We used a composite analysis. To isolate the positive and negative phases of the circulation mode, the index of a specific mode was ranked and 20% of the length of the series was selected from each side (in our case, 14 values). That is, we selected years with the maximum absolute values of the index for each phase, positive or negative. Next, a comparison was made of the number of days with compound extreme index in the positive and negative mode phases. From the mean value in the positive phase of the mode, the mean value in the negative phase was subtracted and presented as a percentage. We added information to the Method section

Point 22: The correlation coefficient given in Table 2 does not indicate whether it is the regional average of the correlation coefficient or the correlation coefficient after regional average.

Response 22: Correlation coefficients are averaged for the region. This information was added to the title of the table, it is also given in the text of the manuscript (line 261)

Point 23: The methods and data are not well described. In the case of global warming, as you can see in Figure 1, the warm and wet extreme index are robust in the winter and the differences of Warm events (WW and WD) in winter between the positive and negative of NAO and AO are significant.

Response 23: We added a data source to the manuscript, supplemented the methodology for calculating composites.

Point 24: The author does not describe how the data is processed,whether the trend is removed from the original data.

Response 24: We did not remove trends from the data series. The results of our previous article showed that detrending does not lead to significant changes in the number of days with extreme precipitation, as well as the correlation between air temperature and precipitation. The works of other authors show that after detrending the time series of compound extreme indices as well as the circulation modes indices, similar results for the correlation coefficients are obtained (for example, Arsenovic, P.; Tosic, I.; Unkasevic, M. Trends in combined climate indices in Serbia from 1961 to 2010. Meteorol. Atmos. Phys. 2015, 127, 489–498. doi: 10.1007/s00703-015-0380)

Point 25: All of the analysis describes the characteristics of the Figures or the relevant results in the table, but does not analyze the mechanism.

Response 25: In the paper we mainly focused on the changes in compound extremes by statistical analysis. Detailed analysis of the physical mechanism of the variations of compound extremes is beyond the scope of this study. We have added some information describing the mechanism to the Discussion section.

Point 26: line 409. Discussion is too short and mixed with conclusions. It is suggested to separate the summary from the discussion. I advise to expand discussion section containing comparison of the results with other studies.

Response 26: We separate Discussion and Conclusion sections and expand discussion section

Point 27: It is meaningful to obtain the period, phase difference and correlation by wavelet spectrum analysis and coherence analysis, but it should be discussed in the sense of weather and climate.

Response 27: We added information to the Discussion section

Reviewer 2 Report

Comments to authors

Review of ArticleConnection of compound extremes of air temperature and precipitation with atmospheric circulation patterns in Eastern Europe” by Olga Sukhonos and Elena Vyshkvarkova

This article deals with the relationship of the major atmospheric circulation modes that have an impact on extreme air temperature and precipitation of the Atlantic-European region covering the period during the past 60 years.

In my opinion, this article is an interesting and well-presented work. I believe that it has the potential to be published in “Climate” Journal after a minor revision. In my opinion, some points need little improvements and also some clarifications are needed. The authors could take under consideration the suggestions, comments and minor comments listed below.

Abstract:

1.       I recommend the authors refer the data-set that it is used in the analysis also in the Abstract.

Introduction:

Data and Methods

1.       In line 103: Please remove the full stop “.”.

2.       In line 123-124 you said that “The indices are taken from the Climate Explorer website  (https://climexp.knmi.nl).

I am a bit critical about this point. It would be better if you could calculate the indices using data from E-obs 20.0 reanalysis (but this data-set provide data only for Europe). The use of two different datasets for the analysis may induce some uncertainties in your results (due to the use of E-obs 20.0 reanalysis for the analysis and also indices that are provided from Climate Explorer website).

Could you please discuss a bit this point?

3.       In lines 163-165: The authors said that “Composite analysis. To isolate the positive and negative phases of the circulation mode, the index of a specific mode was ranked and 20% of the length of the series was selected from each side (in our case, 14 values).”

Did you test the results if you select from each side a different number of cases, for instance 15%, 30% or 35%? The main findings are still the same?

In this context, is it possible the “outliers” of the indices induce uncertainties regarding the results of this study?

Results

As a general comment, I recommend the authors increase the axis font in figures in order to be clearer for the reader.

1.       In line 199: “52nd” please replace with “52nd 

2.       In pages 7 and 8 you have two times “Figure 2”. Please correct the ‘Figure 2’ in line 273 with “Figure 3”.

3.       In line 289. Please remove the full stop.

Conclusion:

1.       In line 364 please replace “21st century” with “21st century”.

Author Response

Response to Reviewer 2 Comments

Dear Reviewer!

First of all, we want to thank you the reading our article carefully and for your valuable comments! Below are the answers to your comments

Review of Article “Connection of compound extremes of air temperature and precipitation with atmospheric circulation patterns in Eastern Europe” by Olga Sukhonos and Elena Vyshkvarkova

This article deals with the relationship of the major atmospheric circulation modes that have an impact on extreme air temperature and precipitation of the Atlantic-European region covering the period during the past 60 years.

In my opinion, this article is an interesting and well-presented work. I believe that it has the potential to be published in “Climate” Journal after a minor revision. In my opinion, some points need little improvements and also some clarifications are needed. The authors could take under consideration the suggestions, comments and minor comments listed below.

Point 1: Abstract: I recommend the authors refer the data-set that it is used in the analysis also in the Abstract.

Response 1: Dataset source were added to the Abstract

Introduction:

Data and Methods

Point 2: In line 103: Please remove the full stop “.”.

Response 2: Corrected

Point 3: In line 123-124 you said that “The indices are taken from the Climate Explorer website  (https://climexp.knmi.nl).

I am a bit critical about this point. It would be better if you could calculate the indices using data from E-obs 20.0 reanalysis (but this data-set provide data only for Europe). The use of two different datasets for the analysis may induce some uncertainties in your results (due to the use of E-obs 20.0 reanalysis for the analysis and also indices that are provided from Climate Explorer website).

Could you please discuss a bit this point?

Response 3: There is not enough data to calculate the circulation modes indices in the E-obs 20.0 reanalysis, because, as you indicated, it really covers only the territory of Europe. We do not correlate fields with different grids. The number of days with compound extremes at each node of the grid is correlated with the time-series of the circulation index.

Point 4: In lines 163-165: The authors said that “Composite analysis. To isolate the positive and negative phases of the circulation mode, the index of a specific mode was ranked and 20% of the length of the series was selected from each side (in our case, 14 values).”

Did you test the results if you select from each side a different number of cases, for instance 15%, 30% or 35%? The main findings are still the same?

In this context, is it possible the “outliers” of the indices induce uncertainties regarding the results of this study?

Response 4: Reducing the threshold leads to a decrease in the number of values involved in the calculation of the composite. This leads to a decrease in the statistical significance of the results. The analysis of composites at thresholds of 30 and 35% of the length of the initial series confirms the results obtained for threshold = 20%. In the article, we present the results for a threshold of 20% of the length of the series as the most representative. As an example, the figure shows the difference composites of the CD index in the winter season for samples of 15, 35% and 20% of the length of the series (See attached file).

Results

Point 5: As a general comment, I recommend the authors increase the axis font in figures in order to be clearer for the reader.

Response 5: The font on the figures is enlarged

Point 6: In line 199: “52nd” please replace with “52nd

Response 6: Corrected

Point 7: In pages 7 and 8 you have two times “Figure 2”. Please correct the ‘Figure 2’ in line 273 with “Figure 3”.

Response 7: Corrected

Point 8: In line 289. Please remove the full stop.

Response 8: Corrected

Point 9: Conclusion:

In line 364 please replace “21st century” with “21st century”.

Response 9: Corrected

Round 2

Reviewer 1 Report

After revision, the manuscript has been greatly improved. But there are still some minor problems that need to be modified.

1.     The right side of the figure 2 cannot be seen clearly, please adjust the position of the figure.

2.     There are too many conclusions to be further condensed.

Author Response

Response to Reviewer 1 Comments (Round 2)

Dear Reviewer!

Thanks for your comments that helped to improve the article greatly!

Below are the answers to your comments

After revision, the manuscript has been greatly improved. But there are still some minor problems that need to be modified.

Point 1: The right side of the figure 2 cannot be seen clearly, please adjust the position of the figure.

Response 1: The position of the figure 2 was changed.

Point 2: There are too many conclusions to be further condensed.

Response 2: Conclusions have been compressed
